# The diagnostic function of intravoxel incoherent motion for distinguishing between pilocytic astrocytoma and ependymoma

Nguyen Minh Duc●[1,2,3]*

1 Doctoral Program, Department of Radiology, Hanoi Medical University, Hanoi, Vietnam, 2 Department of Radiology, Pham Ngoc Thach University of Medicine, Ho Chi Minh City, Vietnam, 3 Department of Radiology, Children's Hospital 02, Ho Chi Minh City, Vietnam

* bsnguyenminhduc@pnt.edu.vn

## Abstract

### Introduction

Intravoxel incoherent motion (IVIM) imaging concurrently measures diffusion and perfusion parameters and has potential applications for brain tumor classification. However, the effectiveness of IVIM for the differentiation between pilocytic astrocytoma and ependymoma has not been verified. The aim of this study was to determine the potential diagnostic role of IVIM for the distinction between ependymoma and pilocytic astrocytoma.

### Methods

Between February 2019 and October 2020, 22 children (15 males and 7 females; median age 4 years) with either ependymoma or pilocytic astrocytoma were recruited for this prospective study. IVIM parameters were fitted using 7 b-values (0–1,500 s/mm$^2$), to develop a bi-exponential model. The diffusivity (D), perfusion fraction ($f$), and pseudo diffusivity (D*) were measured in both tumors and the adjacent normal-appearing parenchyma. These IVIM parameters were compared using the Mann-Whitney U test. Receiver operating characteristic (ROC) curve analysis was employed to assess diagnostic performance.

### Results

The median D values for ependymoma and pilocytic astrocytoma were 0.87 and 1.25 × 10$^{-3}$ mm$^2$/s (p < 0.05), respectively, whereas the $f$ values were 0.11% and 0.15% (p < 0.05). The ratios of the median D values for ependymoma and pilocytic astrocytoma relative to the median D values for the adjacent, normal-appearing parenchyma were 1.45 and 2.10 (p < 0.05), respectively. ROC curve analysis found that the D value had the best diagnostic performance for the differentiation between pilocytic astrocytoma and ependymoma, with an area under the ROC curve of 1.

**Data Availability Statement:** All relevant data are within the paper and its Supporting Information files.

**Funding:** The author(s) received no specific funding for this work.

**Competing interests:** The authors have declared that no competing interests exist.

**Abbreviations:** D, Diffusivity; *f*, Perfusion fraction; D*, Pseudo-diffusivity; IVIM, intravoxel incoherent motion; ROC, receiver operating characteristic; WHO, World Health Organization.

## Conclusion

IVIM is a beneficial, effective, non-invasive, and endogenous-contrast imaging technique. The D value derived from IVIM was the most essential factor for differentiating ependymoma from pilocytic astrocytoma.

## Introduction

After leukemia, pediatric central nervous system tumors represent the second most prevalent cancer type, accounting for approximately 25% of all primary pediatric tumors. Cranial posterior fossa tumors constitute up to 70% of all pediatric brain tumors, and two of the most prominent brain tumors that occur in the posterior cranial fossa are ependymoma and pilocytic astrocytoma [1–3]. The definitive diagnosis of pediatric brain tumors is primarily performed based on histopathology after surgical tumor excision. Occasionally, diagnosis is based on preoperative biopsy, although this method is associated with morbidity and mortality risks [4]. Reliable preoperative, non-invasive, diagnostic imaging is widely accepted to be beneficial for determining the appropriate therapeutic approach. However, diagnostic imaging remains challenging due to tumor diversity and heterogeneity. In up to 10% of cases, pilocytic astrocytoma may present as a superior solid tumor [5]. Because the radiological features compatible with pilocytic astrocytoma are not always coherent, conventional imaging modalities cannot always be used to reach an exact diagnosis [5,6]. Kasliwal et al. reported a case in which the magnetic resonance imaging (MRI) findings supported the diagnosis of an ependymoma-consistent posterior fossa tumor that was confirmed to be pilocytic astrocytoma based on postoperative histopathology [7]. Therefore, ependymoma and pilocytic astrocytoma can present overlapping imaging characteristics, increasing the risk of misdiagnosis with subsequent adverse effects for treatment planning and prognosis.

MRI is the global imaging instrument of choice for the examination of pediatric intracranial tumors [8,9]. Several efforts have been made to analyze whether the parameters of diffusion-weighted imaging (DWI) and perfusion-weighted imaging (PWI) can be exploited for the classification of brain tumors in adults and children [10–12]. In clinical practice, the apparent diffusion coefficient (ADC) is typically measured using 2 b-values (0 and 1,000 s/mm$^2$), which are considered valuable biomarkers for cellular density [13]. Various perfusion parameters, including cerebral blood volume (CBV), cerebral blood flow (CBF), relative enhancement, and the volume transfer constant (K$^{trans}$), are used as major indicators for tumoral vascularity [14].

An imaging technique known as intravoxel incoherent motion (IVIM) was suggested by Le Bihan et al. to represent the molecular motion of water in tissues and could be utilized to distinguish between various diffusion and perfusion aspects [15–17]. Capillary microcirculation may make cohesive microperfusion phases with small b-values challenging to distinguish by ADC [15–17]. Therefore, diffusivity (D) may be impacted by perfusion, resulting in the miscalculation D values, especially in lesions with high perfusion levels. IVIM imaging fits the signal decay to a bi-exponential model by manipulating multiple b-values and segregates pseudo-diffusivity (D*) from the capillary perfusion fraction (*f*), allowing diffusion and perfusion markers to be calculated separately from the same imaging sequence. Because D values are based on a model that is not reliant on the effects of perfusion, the state of diffusion can be more accurately depicted than through conventional ADC measurements. IVIM does not require the use of intravenous contrast agents and can, therefore, be administered to children, pregnant women, patients with a previous extreme allergic/anaphylactic reaction to gadolinium-based

contrast agents, and patients with estimated glomerular filtration rates (eGFR) <30 mL/min/ 1.73 m$^2$ [15–17]. The goal of this study was to test the diagnostic ability of IVIM to distinguish between ependymoma and pilocytic astrocytoma in children.

## Materials and methods

### Patient data

The institutional review board of Children's Hospital 02 approved this prospective study (Ref: 352/NĐ2-CĐT). Before study participation, written informed consent was obtained from the legal guardians of all pediatric patients. Overall, 22 pediatric patients with posterior fossa tumors, who underwent head MRI from February 2019 to October 2020, were divided into two groups: group 1 contained 9 patients with confirmed ependymomas, and group 2 contained 13 patients with confirmed pilocytic astrocytomas. The inclusion criteria were as follows: (i) 7 b-values were obtained from DWI performed before tumor treatment at our hospital; (ii) the operation was performed at our hospital; and (iii) the histopathological results by hematoxylin and eosin staining were able to confirm the tumors as either ependymoma or pilocytic astrocytoma. The exclusion criteria were as follows: (i) other posterior fossa tumors; (ii) MRI prior to surgery performed by a different institution; and (iii) prior interventions, such as biopsy, surgery, radiotherapy, or chemotherapy.

### Data acquisition

All patients underwent scanning with a 1.5 Tesla MRI machine (Multiva; Philips, Best, The Netherlands). DWI was performed to obtain 7 b-values (0, 25, 50, 100, 200, 1,000, and 1,500 s/ mm$^2$), in 3 orthogonal directions, using a single-shot echo-planar sequence in the axial plane, with the following parameters: repetition time, shortest; echo time, shortest; flip angle, 90 degrees; section thickness, 5 mm; gap, 1 mm; field of view (FOV), 230 × 230 mm$^2$; matrix, 144 × 90 mm$^2$; number of acquisition, 2; total scan time, 3.43 minutes.

The Advanced Diffusion Analysis (ADA) mode, available in the Philips Intellispace Portal, version 11, was used to analyze regions of interest (ROIs) defined in the tumors and the adjacent, normal-appearing parenchyma and to calculate bi-exponential IVIM parameters, including D, *f*, and D* (Figs 1 and 2) by two pediatric radiologists with 10-year experience. In this study, it is noted that unblinded single-ROI analysis was carried out. The ratios between the tumor and the adjacent parenchyma were defined by dividing the signal intensity of the tumor by the signal intensity of the adjacent, normal-appearing parenchyma. The ratios for the D, *f*, and D* values are referred to as rD, r*f*, and rD*.

### Statistics

Non-normally distributed continuous variables are presented as the median and interquartile range. The Mann-Whitney U test was used to compare the quantitative parameters between two groups. Sensitivity, specificity, and area under the curve (AUC) were established to evaluate the functional parameters for diagnostic discrimination. The diagnostic thresholds for significant parameters were selected according to the Youden index. Statistical analyses were performed using SPSS, version 26 (IBM Corp., New York, United States). Significant differences were defined at $p < 0.05$.

## Results

The male/female ratios of groups 1 and 2 were 7/2 and 8/5, respectively. The median ages of groups 1 and 2 were 3 and 5 years, respectively.

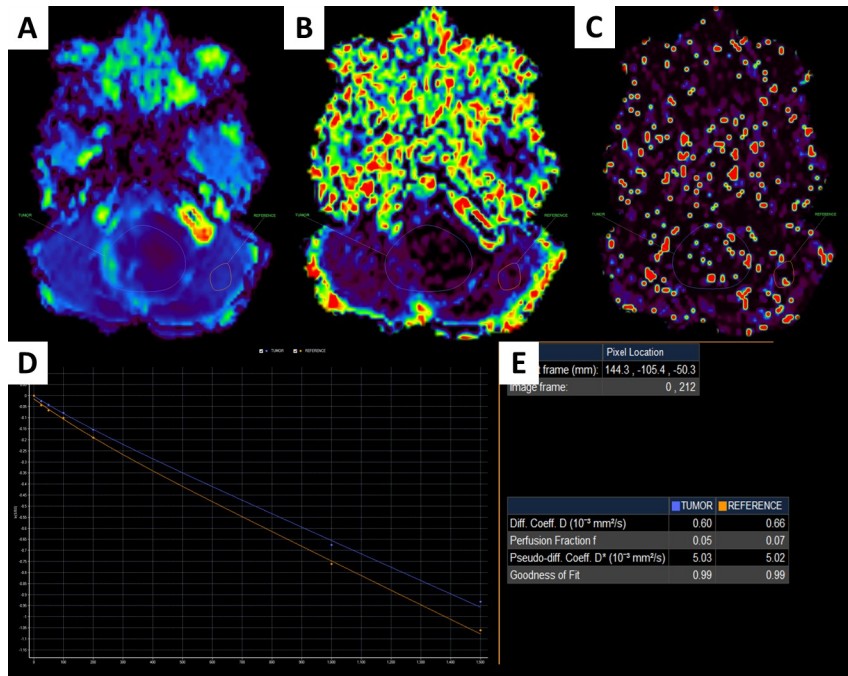

**Fig 1. A 1-year-old male patient with a fourth ventricular tumor, which was verified to be an ependymoma after surgery.** D image (A); $f$ image (B); D* image (C); IVIM signal intensity curve (D); quantitative IVIM parameters of the tumor and the parenchyma (E).

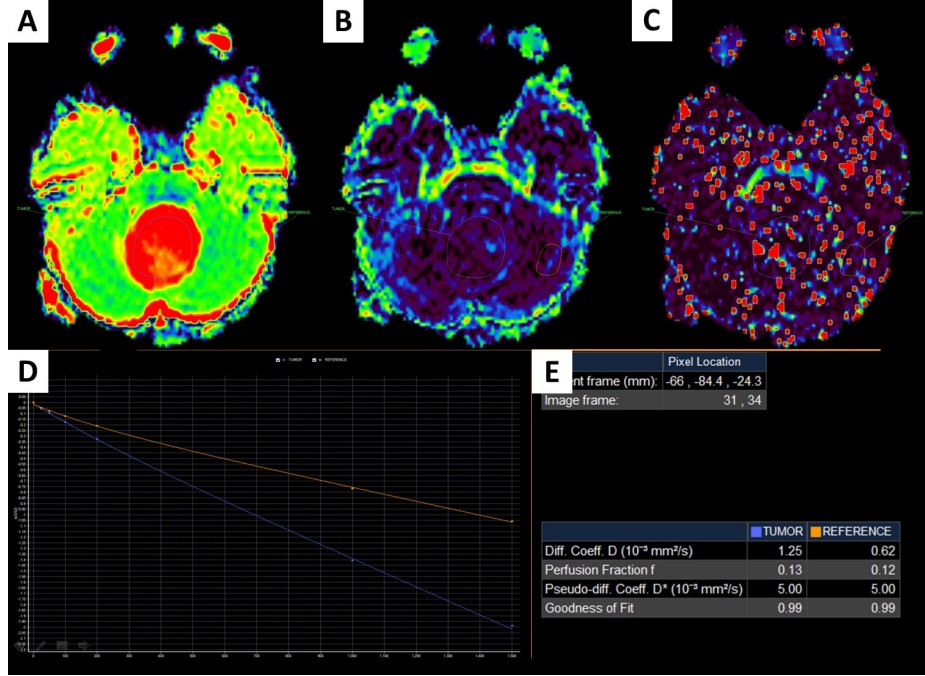

**Fig 2. A 7-year-old female patient with a fourth ventricular tumor, which was verified to be a pilocytic astrocytoma.** D image (A); $f$ image (B); D* image (C); IVIM signal intensity curve (D); quantitative IVIM parameters of the tumor and the parenchyma (E).

**Table 1. Comparison of the IVIM parameters between pilocytic astrocytomas and ependymomas.**

| IVIM parameters | Ependymoma (n = 9) | Pilocytic astrocytoma (n = 13) | p_value |
|---|---|---|---|
| **IVIM** | | | |
| D ($10^{-3}$ mm$^2$/s) | 0.87 (0.34) | 1.25 (0.49) | **< 0.001** |
| $f$ (%) | 0.11 (0.10) | 0.15 (0.08) | **0.030** |
| D* ($10^{-3}$ mm$^2$/s) | 5.00 (0.05) | 5.01 (0.01) | 0.512 |
| **IVIM ratios** | | | |
| rD | 1.45 (0.60) | 2.10 (0.90) | **< 0.001** |
| r$f$ | 2.00 (1.83) | 2.00 (1.62) | 0.556 |
| rD* | 1.00 (0.01) | 1.00 (0.00) | 0.601 |

IVIM: Intravoxel incoherent motion; D: Diffusivity; D*: Pseudo diffusivity; $f$: Perfusion fraction; rD: The ratio between the D value of the tumor and the D value of the parenchyma; rD*: The ratio between the D* value of the tumor and the D* value of the parenchyma; r$f$: The ratio between the $f$ value of the tumor and the $f$ value of the parenchyma.

As described in Table 1, the D, $f$, and rD values for ependymomas were significantly lower than those for pilocytic astrocytomas (p < 0.05).

As illustrated in Table 2, a cut-off D value of 1.030 was established for the precise diagnosis between pilocytic astrocytomas and ependymomas, generating the highest sensitivity value of 100%, a specificity of 100%, and an AUC of 100%. A cut-off rD value of 1.618 was selected for the precise diagnosis between pilocytic astrocytomas and ependymomas, resulting in a sensitivity of 88.9%, a specificity of 100%, and an AUC of 97.4%. A cut-off $f$ value of 0.120 was established for the precise diagnosis between pilocytic astrocytomas and ependymomas, yielding a sensitivity value of 66.7%, a specificity of 84.6%, and an AUC of 77.4% (Fig 3).

## Discussion

Histopathologically, marked differences can be identified between high-grade gliomas (HGG) and low-grade gliomas (LGG), such as on the extent of nuclear malformation, cellular density, and microvascular proliferation [18–27]. The ADC and D values reflect the movement of water molecules and the cellular density inside the tumor [18–27]. Similarly, $f$ values can represent tumoral angiogenesis or microvascular proliferation, which are useful biomarkers for grading brain tumors [18–28]. In the current research, we examined whether preoperative IVIM can be exploited to differentiate ependymoma from pilocytic astrocytoma. We observed important variations between these two tumor types in the parameters D, $f$, and rD. We also defined the most acceptable cut-off values for each proposed parameter that could theoretically be applied in clinical practice for the preoperative separation of ependymoma cases from pilocytic astrocytoma cases. To the best of our knowledge, this is the first study to explore the diagnostic function of IVIM for the differentiation of ependymoma from pilocytic astrocytoma.

**Table 2. ROC analysis illustrating the cut-off value and the AUC, sensitivity, and specificity values, with 95% CIs, for the differential diagnosis between pilocytic astrocytomas and ependymomas.**

| IVIM parameters | Cut-off point | AUC | Sensitivity | Specificity | 95% CI |
|---|---|---|---|---|---|
| D | 1.030 | 1 | 1 | 1 | 1.000–1.000 |
| $f$ | 0.120 | 0.774 | 0.667 | 0.846 | 0.567–0.980 |
| rD | 1.618 | 0.974 | 0.889 | 1 | 0.915–1.000 |

IVIM: Intravoxel incoherent motion; AUC: Area under the curve; CI: Confidence interval; D: Diffusivity; $f$: Perfusion fraction; rD: The ratio between the D value of the tumor and the D value of the parenchyma.

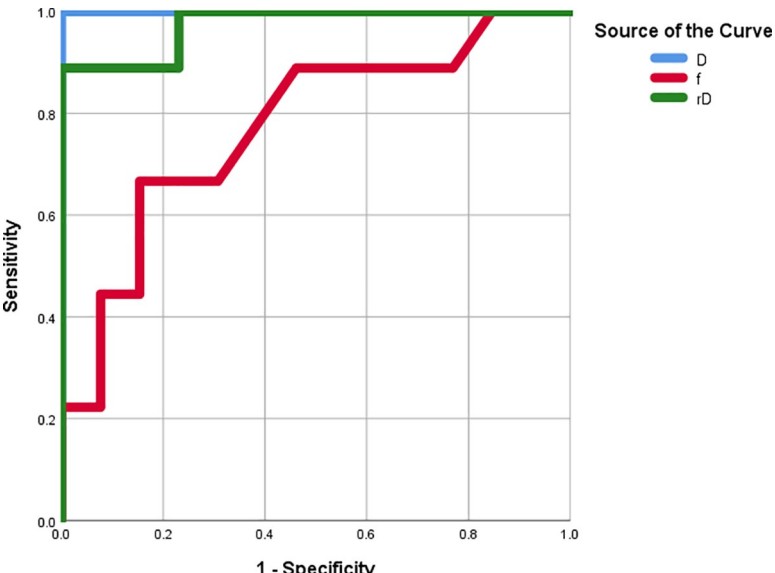

**Fig 3. The ROC curves for the D, *f*, and rD values.**

D values demonstrated perfect sensitivity and specificity for the differentiation of the two tumor types in terms of diffusivity. Previous studies have stated that the diffusion of water molecules is constrained by the histopathological characteristic of malignant tumors, which is primarily due to hypercellularity, expanded nuclei, and high mitotic activity, and contributes to variations in D values between different brain tumor types [18–27]. The quantity of cell membranes in dense tumors results in a decrease in the interstitial space and a decline in the size of the intracellular space, which limits water movement. Thus, D values can be used to evaluate the cell density of brain tumors. The *f* value might reflect the degree of tumor perfusion, which reflects vascularity [18–28]. The degree of vascularity is an important histopathological feature used to classify glioma, as malignant glioma is distinguished from non-malignant glioma by neoangiogenesis [18–28]. In a previous study by Kikuchi et al., the findings showed that *f* values correlated closely with microvessel density (as assessed by anti-CD31 immunostained vascular area/total tissue area ratio), which reflects neoangiogenesis progression [20].

According to the World Health Organization (WHO) classification, pilocytic astrocytoma, which is a benign, grade I tumor, is characterized by Rosenthal fibers, microcystic alterations, and hairy stromal cells or astrocytes. Pilocytic astrocytomas present with hypocellularity, low mitotic activity and rarely undergo malignant transformation. Despite the low cellular density, pilocytic astrocytomas tend to exhibit strong vascularity due to microvascular proliferation and an immensely vascularized stroma [29]. In contrast, ependymoma, which is a predominantly solid tumor, is classified by the WHO classification system as grade II (classic ependymoma) or III (anaplastic ependymoma). Ependymomas present with hypercellularity and high mitotic activity. Fundamentally, decreased water movement, caused by narrow inter- and intracellular spaces, contribute reductions in diffusion signal production. The diffusivity rate in a densely packed tumor, such as ependymoma, is expected to be minimal compared with that in the low-density pilocytic astrocytoma [12]. These factors explain the significantly reduced D, *f*, and rD values observed for ependymomas compared with those for pilocytic astrocytomas (0.87 vs. $1.25 \times 10^{-3}$ mm²/s, 0.11% vs. 0.15%, and 1.45 vs. 2.10, respectively). In some previous studies, the D values for LGG and HGG ranged from $0.52–1.53 \times 10^{-3}$ mm²/s and $0.76–1.02 \times 10^{-3}$ mm²/s, respectively [18–26]. The cut-off D values used to differentiate

LGG from HGG ranged from $1.10–1.25 \times 10^{-3}$ mm$^2$/s. The sensitivity, specificity, and AUC ranged from 81.48%–100%; 56.2%–86.96%; and 74.0%–94.3%, respectively [19,21–23,25,26]. The $f$ values for LGG and HGG ranged from 0.076%–9.5% and 0.1105–21.7%, respectively [18–26]. The cut-off $f$ values for differentiating LGG from HGG ranged from 0.09%–14.1%. The sensitivity, specificity, and AUC values ranged from 77.78%–100%; 78.9%–91.30%; and 82.6%–95.7%, respectively [19,21–23,25,26]. In a recent study by Wang *et al.* [27], the results revealed that rD value of LGG and HDD was 1.811 and 1.425 (p < 0.05), respectively. With the cut-off rD value of 0.828, sensitivity, specificity, and AUC values were 90.5%, 92.3%, and 97.1%, respectively. In this study, the cut-off D and $f$ values for the differentiation between pilocytic astrocytoma and ependymoma were $1.03 \times 10^{-3}$ mm$^2$/s and 0.12%, respectively. We also observed that D values had the best diagnostic performance for the separation of ependymoma cases from pilocytic astrocytoma cases, whereas $f$ values played a moderate role in this process. In a previous study by Dolgorsuren *et al.* [28], the results showed that CBF derived from arterial spin labeling had a better correlation with CBF derived from dynamic susceptibility contrast (DSC) perfusion imaging than $f$ value. In addition, the diffusion coefficient of the fast component on IVIM generated more information on permeability than the $f$ value. Our findings are in line with previous reports [18–28].

This study had some potential limitations. First, the patient population was comparatively small. Second, calculations were based on single ROIs for each case, which may have introduced observer bias. However, single-ROI analysis is commonly used in clinical practice. Whole-tumor segmentation could resolve this problem with histogram analysis. Third, for diffusion encoding, we only used 3 perpendicular directions. Anisotropic analysis through the diffusion tensor imaging system involves the use of at least 6 directions in anisotropic tissues; however, DWI averaged over 3 directions is frequently used during the diagnostic process in clinical practice. Fourth, only 2 types of posterior fossa tumors were investigated in this study. Finally, we only focused on the comparison of IVIM parameters between two tumor types without the assessment of the conventional ADC, classical DSC perfusion, and morphological information. To validate our results, further research should be performed examining additional tumor types in larger samples, investigating different measurement techniques, and combining morphological data.

## Conclusion

IVIM allows details regarding tumor diffusivity and vascularity to be obtained concurrently, without depending on the use of intravenous contrast agents, which allows for this method to be applied in children, pregnant women, patients with severely compromised renal function, and patients with severe allergies to contrast agents. The D, $f$, and rD values were useful for tumor type differentiation, with the D value displaying the best diagnostic performance. Despite the moderate performance of the $f$ value, it can contribute partially to the diagnostic differentiation between pilocytic astrocytoma and ependymoma.

## Acknowledgments

I would like to express my gratitude to Dr. Mai Tan Lien Bang, Dr. Dang Do Thanh Can, Dr. Huynh Quang Huy, Mr. Bilgin Keserci, Mrs Dang Thi Bich Ngoc and Mr. Nguyen Chanh Thi for their assistance and technical support in completing this research.

### Ethics statement

The institutional review board of Children's hospital 2 approved this prospective study (Ref: 352/NĐ2-CĐT). Informed consent of authorized guardians of patients was obtained.

## Author Contributions

**Conceptualization:** Nguyen Minh Duc.

**Data curation:** Nguyen Minh Duc.

**Formal analysis:** Nguyen Minh Duc.

**Methodology:** Nguyen Minh Duc.

**Writing – original draft:** Nguyen Minh Duc.

**Writing – review & editing:** Nguyen Minh Duc.

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
