## [Decision Letter · Decision Letter 0]

4 Dec 2020

PONE-D-20-31294

The diagnostic function of intravoxel incoherent motion for distinguishing between pilocytic astrocytoma and ependymoma

PLOS ONE

Dear Dr. Minh Duc,

Thank you for submitting your manuscript to PLOS ONE. After careful consideration, we feel that it has merit but does not fully meet PLOS ONE’s publication criteria as it currently stands. Therefore, we invite you to submit a revised version of the manuscript that addresses the points raised during the review process.

 Please response the questions from the reviewers. Lack of correlation with the classical DSC perfusion.

We look forward to receiving your revised manuscript.

Kind regards,

Quan Jiang, Ph,D.

Academic Editor

PLOS ONE

Journal Requirements:

2.We note that you have indicated that data from this study are available upon request. PLOS only allows data to be available upon request if there are legal or ethical restrictions on sharing data publicly. For information on unacceptable data access restrictions, please see http://journals.plos.org/plosone/s/data-availability#loc-unacceptable-data-access-restrictions.

Reviewers' comments:

Reviewer's Responses to Questions

**Comments to the Author**

1. Is the manuscript technically sound, and do the data support the conclusions?

Reviewer #1: Partly

Reviewer #2: Yes

2. Has the statistical analysis been performed appropriately and rigorously? 

Reviewer #1: Yes

Reviewer #2: Yes

3. Have the authors made all data underlying the findings in their manuscript fully available?

Reviewer #1: Yes

Reviewer #2: Yes

4. Is the manuscript presented in an intelligible fashion and written in standard English?

Reviewer #1: Yes

Reviewer #2: Yes

5. Review Comments to the Author

Reviewer #1: Introduction:

- References [1-3] are cited three times. Please cite them only once.

- Line 15: coonfirmed o be==> confirmed to be

- You say IVIM was "recently" described- It was described in the 80s, so please use other expression.

Data acquisition:

- How many different observers perfomed the measures? How much experience in pediatric Neurorradiology did they have?

Was it unblinded or blinded?

- How many ROIs were used?

- Please describe the methods for hystopathologic diagnosis

Results:

- Group 1 and 2: Please define what group was each type of tumor.

- In order to analyze the advantages of IVIM measurements compared to teh use of ADC it would be interesting to compare values of ADC in both type of tumors too in this study.

Discussion

Line 25:It would be intersting to present hystopathological data of anti CD31 in this study

Reviewer #2: In this study, the authors have analyzed the IVIM imaging for the diagnostic's distinction between ependymoma and pilocytic astrocytoma. The results are quite clear demonstrating a significant D and f values between these two tumors. The D value derived from IVIM is the most essential factor. The IVIM sequence is interesting because it makes it possible to dispense with the injection of a contrast agent in order to estimate tumoral perfusion. The interest of this study is its application on tumors not explored in previous studies which are focused on metastasis, low and high grade glioma, meningioma and lymphoma. The limits of this study are the small numbers of patients and the lack of correlation with the classical DSC perfusion (which has been done in other glioma studies).

I have some comments about this article:

1) This study was conducted on a 1.5T. Are there any studies comparing IVIM sequence between magnetic fields?

2) The lesion characteristics are not detailed, in particular the tumor sizes which may participate to partial volume effect.

3) The authors used the Philips software, would it be possible to have the estimation model: classic monoexponential or biexponential?

4) Some recent references are missing:

Ki-67 labeling index and the grading of cerebral gliomas by using intravoxel incoherent motion diffusion-weighted imaging and three-dimensional arterial spin labeling magnetic resonance imaging. Chaochao Wang , Haibo Dong Acta Radiol . 2020 Aug;61(8):1057-1063.

Correlation and Characteristics of Intravoxel Incoherent Motion and Arterial Spin Labeling Techniques Versus Multiple Parameters Obtained on Dynamic Susceptibility Contrast Perfusion MRI for Brain Tumors. Enkh-Amgalan olgorsuren, Masafumi Harada, Yuki Kanazawa , Takashi Abe, Maki Otomo, Yuki Matsumoto, Yoshifumi Mizobuchi, Kohhei Nakajima. J. Med. Invest.. 2019;66(3.4):308-313. The conclusion of this study is that compared to DSC, the ASL-CBF is more suitable for the evaluation of perfusion in brain tumors than IVIM parameters and should be discussed.

6. PLOS authors have the option to publish the peer review history of their article (what does this mean?). If published, this will include your full peer review and any attached files.

Reviewer #1: **Yes: **Nerea Domínguez-Pinilla

Reviewer #2: No

---

## [Author Response · Author response to Decision Letter 0]

21 Jan 2021

December 8, 2020 

Dear Editor 

Thank you for giving us the opportunity to improve and resubmit our manuscript “The diagnostic function of intravoxel incoherent motion for distinguishing between pilocytic astrocytoma and ependymoma” Ms. PONE-D-20-31294

Please find enclosed the revised manuscript for further consideration. The manuscript has been revised according to the comments raised by the reviewer to the best of our ability. Please find a detailed reply to the reviewer comments attached to this revision. We would like to thank the reviewers for the constructive and competent criticism, and we hope that our manuscript will be acceptable for publication in your journal.

I look forward to hearing from you.

Best regards, 

Nguyen Minh Duc

Department of Radiology - Pham Ngoc Thach University of Medicine

Email: bsnguyenminhduc@pnt.edu.vn

Reviewer #1: 

- References [1-3] are cited three times. Please cite them only once.

+ We are grateful to reviewer’ comment. We have revised text accordingly as requested. Please refer to page 4 lines from 2 to 6. We hope our revision makes make reviewer satisfactory.

- Line 15: coonfirmed o be==> confirmed to be

+ We are grateful to reviewer’ comment. We have revised text accordingly as requested. Please refer to page 4 line 16. We hope our revision makes make reviewer satisfactory.

- You say IVIM was "recently" described- It was described in the 80s, so please use other expression.

+ We are grateful to reviewer’ comment. We have revised text accordingly as requested. Please refer to page 4 line 28. We hope our revision makes make reviewer satisfactory.

Data acquisition:

- How many different observers perfomed the measures? How much experience in pediatric Neurorradiology did they have?

Was it unblinded or blinded?

How many ROIs were used?

+ We are grateful to reviewer’ comment. We have revised text accordingly as requested to clarify these points. Please refer to page 6 lines from 5 to 7 and page 10 lines from 6 to 7. We hope our revision makes make reviewer satisfactory. 

- Please describe the methods for hystopathologic diagnosis

+ We are grateful to reviewer’ comment. We have revised text accordingly as requested to clarify these points. Please refer to page 5 lines from 22 to 23. We hope our revision makes make reviewer satisfactory. 

- Group 1 and 2: Please define what group was each type of tumor.

+ We are grateful to reviewer’ comment. We have revised text accordingly as requested to clarify these points. Please refer to page 5 lines from 18 to 20. We hope our revision makes make reviewer satisfactory. 

- In order to analyze the advantages of IVIM measurements compared to teh use of ADC it would be interesting to compare values of ADC in both type of tumors too in this study.

+ We are grateful to reviewer’ comment. In this study, our aim was to perform only IVIM analysis. However, reviewer’s recommendation is appropriate; thus, we have added this to the limitation part of this study. Please refer to page 10 lines from 12 to 16. We hope our revision makes make reviewer satisfactory. 

Discussion

Line 25:It would be intersting to present hystopathological data of anti CD31 in this study

+ We are grateful to reviewer’ comment. For confirming ependymoma and pilocytic astrocytoma, H&E staining can be resolved efficaciously and anti CD31 was not needed. Thus we are very sorry to reviewer that this request cannot be done. We hope reviewer finds our explanation appropriate.

Reviewer #2: 

In this study, the authors have analyzed the IVIM imaging for the diagnostic's distinction between ependymoma and pilocytic astrocytoma. The results are quite clear demonstrating a significant D and f values between these two tumors. The D value derived from IVIM is the most essential factor. The IVIM sequence is interesting because it makes it possible to dispense with the injection of a contrast agent in order to estimate tumoral perfusion. The interest of this study is its application on tumors not explored in previous studies which are focused on metastasis, low and high grade glioma, meningioma and lymphoma. The limits of this study are the small numbers of patients and the lack of correlation with the classical DSC perfusion (which has been done in other glioma studies).

+ We are grateful to reviewer’ comment. We have revised text accordingly as requested. Please refer to page 10 lines from 12 to 16. We hope that reviewer find our revision satisfactory.

1) This study was conducted on a 1.5T. Are there any studies comparing IVIM sequence between magnetic fields?

+ Up to now, IVIM in each study was performed on 1.5 or 3T. The information of IVIM between two or more magnetic fields was not investigated yet. We hope that our explanation makes reviewer feel satisfactory.

2) The lesion characteristics are not detailed, in particular the tumor sizes which may participate to partial volume effect.

+ We are grateful to reviewer’ comment. We have revised text accordingly as requested. Please refer to page 10 lines from 5 to 16. We hope that reviewer find our revision satisfactory.

3) The authors used the Philips software, would it be possible to have the estimation model: classic monoexponential or biexponential?

+ We are grateful to reviewer’ comment. We have revised text accordingly as requested. Please refer to page 6 lines from 5 to 7. We hope that reviewer find our revision satisfactory.

4) Some recent references are missing:

Ki-67 labeling index and the grading of cerebral gliomas by using intravoxel incoherent motion diffusion-weighted imaging and three-dimensional arterial spin labeling magnetic resonance imaging. Chaochao Wang , Haibo Dong Acta Radiol . 2020 Aug;61(8):1057-1063.

Correlation and Characteristics of Intravoxel Incoherent Motion and Arterial Spin Labeling Techniques Versus Multiple Parameters Obtained on Dynamic Susceptibility Contrast Perfusion MRI for Brain Tumors. Enkh-Amgalan olgorsuren, Masafumi Harada, Yuki Kanazawa , Takashi Abe, Maki Otomo, Yuki Matsumoto, Yoshifumi Mizobuchi, Kohhei Nakajima. J. Med. Invest.. 2019;66(3.4):308-313. The conclusion of this study is that compared to DSC, the ASL-CBF is more suitable for the evaluation of perfusion in brain tumors than IVIM parameters and should be discussed.

+ We are grateful to reviewer’ comment. We have revised text accordingly and added the information in these studies as requested. Please refer to page 9 lines from 24 to 27 and page 10 lines from 1 to 3. We also updated the order of references based on adding two new references of 27, 28 as suggested. We hope that reviewer finds our revision satisfactory.

---

## [Decision Letter · Decision Letter 1]

17 Feb 2021

The diagnostic function of intravoxel incoherent motion for distinguishing between pilocytic astrocytoma and ependymoma

PONE-D-20-31294R1

Dear Dr. Minh Duc,

We’re pleased to inform you that your manuscript has been judged scientifically suitable for publication and will be formally accepted for publication once it meets all outstanding technical requirements.

Kind regards,

Quan Jiang, Ph,D.

Academic Editor

PLOS ONE

Additional Editor Comments (optional):

Reviewers' comments:

Reviewer's Responses to Questions

**Comments to the Author**

1. If the authors have adequately addressed your comments raised in a previous round of review and you feel that this manuscript is now acceptable for publication, you may indicate that here to bypass the “Comments to the Author” section, enter your conflict of interest statement in the “Confidential to Editor” section, and submit your "Accept" recommendation.

Reviewer #1: All comments have been addressed

2. Is the manuscript technically sound, and do the data support the conclusions?

Reviewer #1: Yes

3. Has the statistical analysis been performed appropriately and rigorously? 

Reviewer #1: Yes

4. Have the authors made all data underlying the findings in their manuscript fully available?

Reviewer #1: Yes

5. Is the manuscript presented in an intelligible fashion and written in standard English?

Reviewer #1: Yes

6. Review Comments to the Author

Reviewer #1: (No Response)

7. PLOS authors have the option to publish the peer review history of their article (what does this mean?). If published, this will include your full peer review and any attached files.

Reviewer #1: **Yes: **Nerea Domínguez-Pinilla

---

## [Editor Report · Acceptance letter]

19 Feb 2021

PONE-D-20-31294R1 

The diagnostic function of intravoxel incoherent motion for distinguishing between pilocytic astrocytoma and ependymoma 

Dear Dr. Minh Duc:

I'm pleased to inform you that your manuscript has been deemed suitable for publication in PLOS ONE. Congratulations! Your manuscript is now with our production department. 

Kind regards, 

on behalf of

Dr. Quan Jiang 

Academic Editor

PLOS ONE